

**PeerJ Hubs**
Published on behalf of

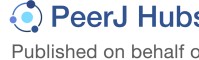

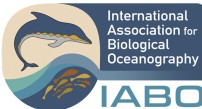

# Analysis of imaging data on seasonal changes in the population structure and vertical distribution of two dominant planktonic copepod species in the western subarctic Pacific

Tian Gao[1] and Atsushi Yamaguchi[1,2]

[1] Graduate School of Fisheries Sciences, Hokkaido University, Hakodate, Hokkaido, Japan
[2] Arctic Research Center, Hokkaido University, Sapporo, Hokkaido, Japan

Corresponding author
Atsushi Yamaguchi,
a-yama@fish.hokudai.ac.jp

## ABSTRACT

Traditionally, zooplankton analyses have relied on stereomicroscopes, but recent advancements in imaging analysis have offered significant advantages, including the simultaneous collection of abundance, size, and biovolume data. In this study, formalin-preserved samples were collected from depths of 0 to 1,000 m across four seasons at a station in the western subarctic Pacific, using the imaging device ZooScan. Two dominant copepod genera, *Metridia* and *Eucalanus*, were examined for seasonal changes in abundance, biovolume, and diurnal vertical distribution. ZooScan measurements were taken for each developmental stage to obtain information on the equivalent spherical diameter (ESD). Four *Metridia* species were identified: *M. pacifica*, *M. okhotensis*, *M. asymmetrica*, and *M. curticauda*. *M. pacifica*, the dominant species, had an ESD of 2 mm or less, while the other three species exceeded 2 mm. *M. pacifica* exhibited diurnal migration to surface layers (0–50 m) at night, while the larger species were primarily located in the deeper layer (750–1,000 m) both day and night. Only one species, *E. bungii*, was identified in the genus *Eucalanus*, with size cohorts corresponding to each developmental stage. Its vertical distribution was consistent day and night across seasons, but seasonal changes were evident. In October and February, *E. bungii* was found at depths of 200–500 m. In April, later developmental stages migrated to shallower depths of 50–200 m, while in July, younger stages (C1–C4) were found at 0–50 m, indicating recent reproduction during the spring phytoplankton bloom. Although it was clear that new individuals emerged in July, understanding the dynamics of later stages and generation time was difficult due to overlapping size distributions, particularly in C5 and C6. The differences in vertical distribution between copepod species reveal important ecological trends: *M. pacifica* performs diel vertical migration (DVM), while *E. bungii* exhibits seasonal vertical migration (SVM). *M. pacifica* shows no clear seasonality in population structure, whereas *E. bungii* has distinct seasonal patterns. This indicates that both species reproduce near the surface, but *E. bungii* follows annual life cycles due to its larger size, while *M. pacifica* reproduces opportunistically throughout the year. Using ZooScan offers significant advantages for studying copepod ecology, enabling precise estimations of ecological fluxes—such as feeding, production, and egestion—through accurate measurements of body sizes

and masses. Adopting these methods will enhance our understanding of copepod populations and their ecosystems.

## INTRODUCTION

In the western subarctic Pacific, St. K2 has been established as a long-term time-series observation station by Japan Agency for Marine-Earth Science and Technology (JAMSTEC). The quantification and flux of various biogeochemical parameters from this station have been reported (*Honda et al., 2017*). Research on zooplankton at St. K2 has provided several key findings, including estimates of vertical carbon flux to the deep layers achieved through the ontogenetic vertical migration of large-sized copepods as they enter diapause (*Kobari et al., 2008*). Other studies have compared zooplankton communities at St. K2 with those at subtropical stations (*Steinberg et al., 2008a*) and estimated the consumption of sinking particles by mesozooplankton and heterotrophic bacteria in the deep layers (*Steinberg et al., 2008b*). Additionally, the community structure of zooplankton at St. K2 has been analyzed in comparison to subtropical stations (*Kitamura et al., 2016*), and seasonal changes in zooplankton collected from time-series sediment traps at a depth of 200 m have been documented (*Yokoi et al., 2018*). While these findings are significant, most have focused on elucidating community structure and material circulation, leaving a gap in our understanding of the dynamics and population structure of dominant species within each zooplankton taxonomic group. To address this gap, examples of species-level analyses have been conducted using vertically stratified samples collected both during the day and night across four seasons. *Amei et al. (2021)* analyzed pelagic polychaetes, while *Taniguchi et al. (2023)* focused on pelagic amphipods. Additionally, two other studies have documented seasonal changes in population structure, including body size and gonadal development of dominant species, such as the chaetognath *Eukrohnia hamata* (*Nakamura, Zhang & Yamaguchi, 2023*) and the hydromedusa *Aglantha digitale* (*Aizawa, Gao & Yamaguchi, 2023*).

Recent ecological information on certain macrozooplankton taxa has been accumulated (see above). However, to understand material flux through marine zooplankton adequately, we particularly need information on the basic taxon: copepods (*Kobari et al., 2008*). Copepods are the most dominant group in terms of both abundance and biomass within the zooplankton community of the western subarctic Pacific (*Ikeda, Shiga & Yamaguchi, 2008*). The large-sized copepod genus *Neocalanus* is particularly notable for its biomass, comprising three sympatric species: *N. cristatus, N. flemingeri,* and *N. plumchrus* (*Kobari & Ikeda, 2000*). Identifying these three species requires the use of a stereomicroscope, making it challenging to discern them from imaging data obtained through other instruments. For this reason, we did not target *Neocalanus* spp. in this study. In addition to *Neocalanus, Metridia pacifica* and *Eucalanus bungii* are two copepod species that also have significant

biomass and large body sizes (*Ikeda, Shiga & Yamaguchi, 2008*). The life cycles of these two species have been documented in the Oyashio region of the western North Pacific subarctic (*Padmavati, Ikeda & Yamaguchi, 2004*; *Shoden, Ikeda & Yamaguchi, 2005*) and in the Gulf of Alaska in the eastern North Pacific subarctic (*Miller et al., 1984*; *Batchelder, 1985*). In terms of ecological characteristics, *M. pacifica* and *E. bungii* feed near the surface to gather energy for reproduction, making them income breeders (*Yamaguchi et al., 2010a*). In contrast, the biomass-dominant *Neocalanus* species only feed during the juvenile stages in the surface layer before sinking to deeper waters, where they breed without feeding, relying on stored energy for maturation and reproduction (*Kobari & Ikeda, 2000*). Therefore, *Neocalanus* species are considered capital breeders. The differences in reproductive strategies between the dominant copepod species suggest that *M. pacifica* and *E. bungii* are more flexible and better adapted to environmental changes. From this perspective, studying these species is crucial for predicting future changes in the ecosystem.

Recent studies on zooplankton at St. K2 have shifted from traditional stereomicroscopes to advanced imaging techniques. A standout innovation is ZooScan, developed in the 2000s, which measures individual zooplankton size and identifies their species from images, proving effective in various marine ecosystems (*Gorsky et al., 2010*; *Irisson et al., 2022*). At St. K2, ZooScan is used particularly for analyzing pelagic amphipods (*Taniguchi et al., 2023*). In their study, *Taniguchi et al. (2023)* utilized ZooScan on amphipod samples sorted with a stereomicroscope, enabling detailed assessments of population structure in terms of both abundance and biovolume. For the dominant species, *Themisto pacifica*, they analyzed growth and dynamics based on the equivalent spherical diameter (ESD). The advantages of ZooScan for in-depth analysis of specific planktonic taxa are increasingly clear and could enhance our understanding of marine biodiversity.

In this study, we conducted ZooScan measurements on formalin-preserved zooplankton samples collected from vertical stratified sites at depths ranging from 0 to 1,000 m at Station K2 during four consecutive seasons. We identified two large copepod species: *Metridia* spp. and *E. bungii*, based on the images obtained. Our analysis focused on seasonal changes in the abundance and vertical distribution of the biovolume of these species. Additionally, we sorted *M. pacifica* and *E. bungii* from the formalin-preserved samples according to each copepodite stage. ZooScan measurements were performed on each stage to assess the ESD of each stage. This study aims to clarify the species-specific differences in the vertical distribution and population dynamics of two income-breeding copepods: *M. pacifica* and *E. bungii*. Additionally, we discuss the advantages and limitations of using the ZooScan imaging device in ecological studies of specific zooplankton species. Future applications of the ZooScan imaging device in ecological research on these specific species will also be explored.

## MATERIAL AND METHODS

### Field sampling

On October 29, 2010, February 26, April 22, and July 3–4, 2011, we conducted day and night vertically stratified oblique tows using a multi-stage open-close net system called

**Table 1  Sampling data at St. K2 in the western subarctic Pacific during October 2010 to July 2021.** All the samples were collected from eight discrete depths between 0 and 1,000 m (0–50, 50–100, 100–150, 150–200, 200–300, 300–500, 500–750, 750–1,000 m) by oblique tow of IONESS.

| Sampling date | Local time | D/N |
|---|---|---|
| 29 Oct. 2010 | 12:09–13:52 | D |
| 29 Oct. 2010 | 22:09–23:38 | N |
| 26 Feb. 2011 | 12:35–14:41 | D |
| 26 Feb. 2011 | 22:01–23:44 | N |
| 22 Apr. 2011 | 21:59–23:56 | N |
| 22 Apr. 2011 | 12:45–14:37 | D |
| 3 July 2011 | 12:05–13:55 | D |
| 3/4 July 2011 | 22:51–0:55 | N |

Intelligent Operative Net Sampling System (IONESS, SEA Co. Ltd.) with a mesh size of 335 µm and an opening area of 1.5 m². A flowmeter was installed to measure the filtered volume, and no clogging occurred. The net was divided into eight layers corresponding to depths of 0–50 m, 50–100 m, 100–150 m, 150–200 m, 200–300 m, 300–500 m, 500–750 m, and 750–1,000 m, at Station K2 (47°N, 160°E; depth 5,230 m) in the western subarctic Pacific (see Table 1, Fig. 1). We defined "day" and "night" based on the sunrise and sunset times for each sampling date. To minimize the effects of transitional periods, we avoided sampling two hours before and after sunrise and sunset. The collected samples were immediately preserved on board the ship using 5% borax-buffered formalin in seawater. During each sampling period, environmental data, including water temperature, salinity, dissolved oxygen concentration, and fluorescence, were measured with a CTD (SBE 911 plus; Sea-Bird Electronics Inc.).

## ZooScan measurements

In the land laboratory, the zooplankton samples were divided into subsamples ranging from 1/2 to 1/128, depending on the abundance of the zooplankton. Measurements were conducted using the ZooScan imaging device (ZooScan MIII, Hydroptic Inc.), following the methodologies outlined by *Gorsky et al. (2010)*. Initially, the ZooScan scanning cell was filled with distilled water, and a background scan was performed. Next, the zooplankton sample was poured into the scanning cell, and the sample was scanned. During the scanning process, care was taken to adjust the sample's position using a dissection needle or other tools. This was done to prevent the sample from floating on the water surface or to avoid overlap between multiple individual organisms. The scanned images were then segmented into individual images using ZooProcess in ImageJ software. Finally, these images were uploaded to EcoTaxa (http://ecotaxa.obs-vlfr.fr/) *via* FileZilla to acquire species identification and equivalent spherical diameter (ESD, measured in mm) data for each individual. We manually identified each captured image using ZooScan. We determined the area excluded in square millimeters (mm²) from the area excluded data in pixels by using a conversion factor of 10.58 µm per pixel (*Gorsky et al., 2010*). The equivalent

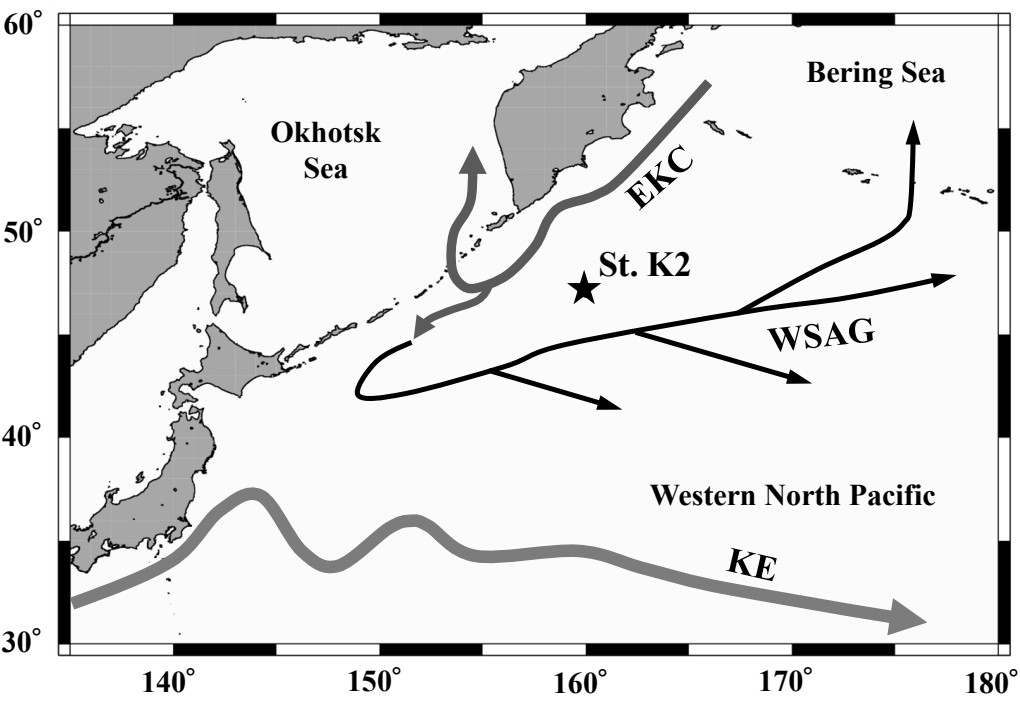

**Figure 1** **Location of the sampling station (K2) in the western subarctic Pacific gyre.** Arrows represent approximate positions and directions of the currents (*cf. Yasuda, 2003*). EKC: East Kamchatka Current, WSAG: Western Subarctic Gyre, KE: Kuroshio Extension.

spherical diameter (ESD) was then calculated using the formula:

$$ESD\ (mm) = 2 \times \sqrt{\frac{Area\ excluded\ (mm^2)}{\pi}}.$$

The biovolume (mm³) for each image can be calculated using the formula:

$$Biovolume\ (mm^3) = \frac{4}{3}\pi\left(\frac{ESD\,(mm)}{2}\right)^3.$$

## Data analysis

*Metridia* spp. and *E. bungii* dominate the copepod community in terms of both abundance and biomass, and their distinct morphological features make them easy to identify from other species. The *Metridia* genus includes *M. pacifica*, *M. okhotensis*, *M. asymmetrica*, and *M. curticauda* (Fig. 2). For each sample, we calculated the abundance (ind. m$^{-3}$) and biovolume (mm³ m$^{-3}$) of both species, as well as the ESD, based on the aliquot rate and filtered volume. These abundance and biovolume values were then multiplied by the net towing depth (in meters) to determine the abundance (ind. m$^{-2}$) and biovolume (mm³ m$^{-2}$) per square meter, providing estimates for each depth layer in the water column. To assess the abundance in the water column, we integrated the abundance (ind. m$^{-2}$) across all layers of the 0–1,000 m water column. The abundance percentage at each layer

was calculated as a proportion of the total abundance in the water column, enabling us to evaluate the vertical distribution of the population. We created histograms in 0.1 mm ESD intervals to analyze the population structure based on the abundance of values within the 0–1,000 m water column. Additionally, we examined the vertical distribution by assessing the percentage of each depth layer relative to the total abundance in the water column, using 0.1 mm ESD intervals once again.

While all the samples were scanned without a sorting state, to identify each species and stage, individuals of *M. pacifica* and *E. bungii* found in the IONESS samples were sorted by copepodite stage from their most abundant samples. ZooScan measurements were then taken for each copepodite stage. The mean and standard deviation of the ESD were calculated for each stage. In contrast, for *M. okhotensis*, *M. asymmetrica*, and *M. curticauda*, ZooScan measurements were conducted solely on adult females (C6F), and the mean and standard deviation of ESD were also calculated for these species.

## RESULTS

### Hydrological environments

Figure 3 displays the vertical distributions of water temperature, salinity, dissolved oxygen, and fluorescence for each sampling period. Throughout the study, water temperatures ranged from 0.7 to 8.5 °C, salinity varied between 32.5 and 34.5, dissolved oxygen concentrations ranged from 0.3 to 7.5 ml L$^{-1}$, and fluorescence levels ranged from 0.02 to 2.32. A seasonal thermocline was observed at around 50 m in October and June, while in February and April, the temperature remained nearly uniform down to 100 m. Across all sampling periods, the water temperature exhibited a minimum of 1–2 °C at depths of 100 m, peaked at approximately 3.5 °C around 200 m, and then declined with increasing depth below 200 m. Salinity consistently increased with depth during all seasons, although it was uniform above 100 m in February and April. In June and October, salinity remained low (below 33) at depths above 50 m. Dissolved oxygen (DO) levels decreased with increasing depth; although some concentrations were measured down to 200 m, they dropped to extremely low levels (below 2 ml L$^{-1}$) at depths greater than 200 m. Fluorescence was predominantly high only above the thermocline during all seasons, with the peak value recorded in June.

### *Metridia* spp.

Figure 4 displays the day and night abundance (ind. m$^{-2}$) and biovolume (mm$^3$ m$^{-2}$) of *Metridia* copepods, along with their abundance distribution across eight layers in the 0–1,000 m water column, categorized by a 0.1 mm ESD interval. Among the four *Metridia* species studied, *M. pacifica* was smaller, with an ESD below two mm. In contrast, the other three *Metridia* species had an ESD greater than two mm (Fig. 4). As a guide for identification, we displayed the ESD data for each species and stage at the bottom of the panel (Figs. 4 and 5). During the day, *M. pacifica* specimens, particularly the larger individuals, were found at greater depths. However, they exhibited a distinct diel vertical migration, rising to the surface layer (0–50 m) at night across all four seasons. It should also be noted that *M. pacifica* C6M stayed in the deep layer both day and night.

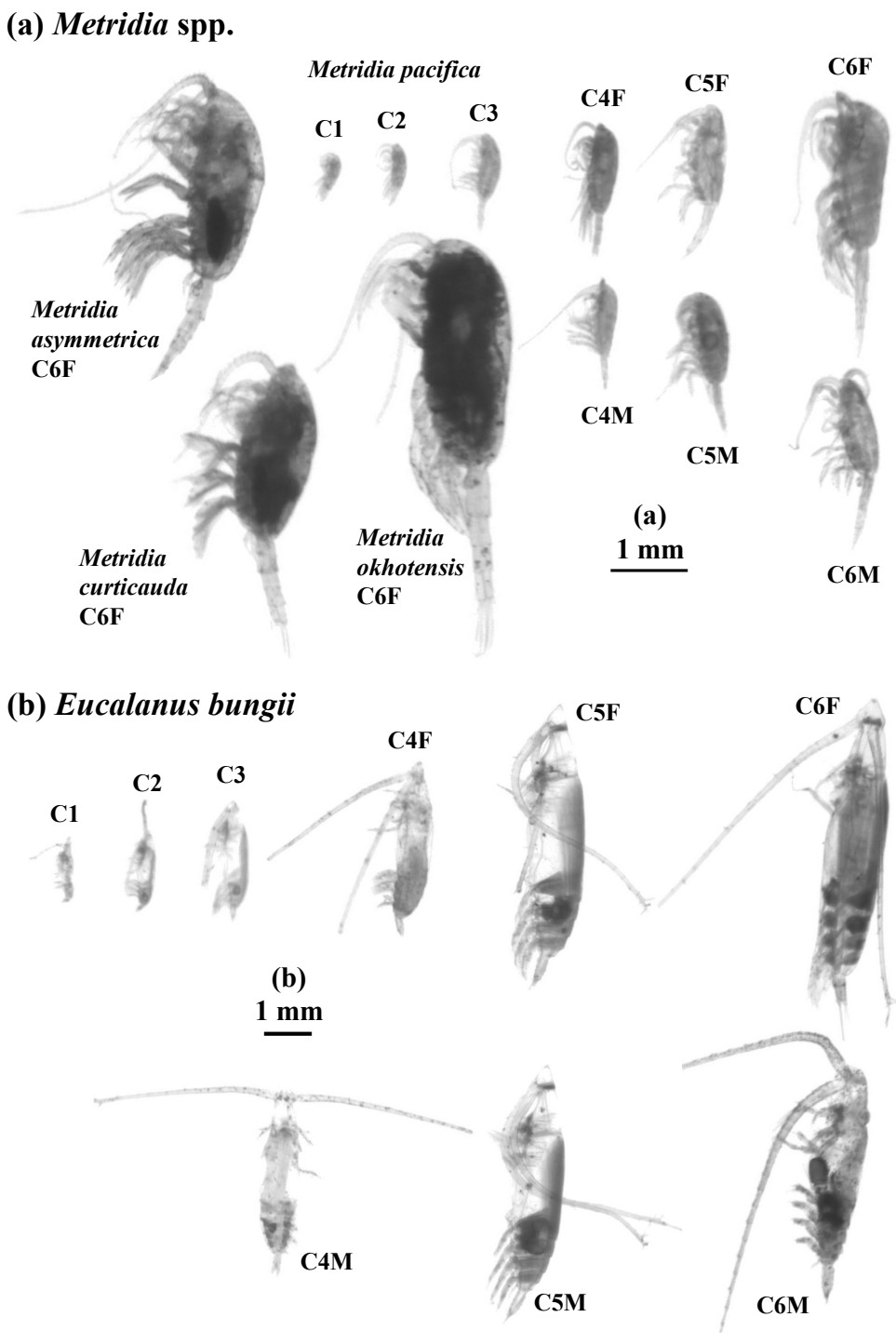

**Figure 2** ZooScan images of two treated general/species in this study: *Metridia* spp. (a) and *Eucalanus bungii* (b). All copepodite stages were captured for *M. pacifica* and *E. bungii* . Note that scale bars are varied between (a) and (b). F: female, M: male.

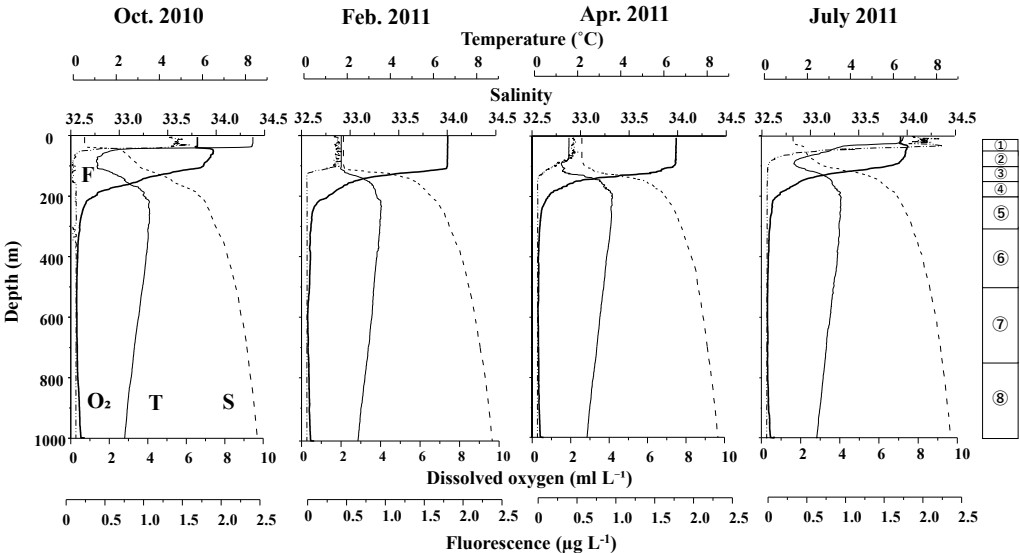

**Figure 3** Vertical changes in temperature (T), salinity (S), dissolved oxygen (O₂) and fluorescence (F) at St. K2 in the western subarctic Pacific from October 2010 to June 2011. Circled numbers in the right column indicate the depth strata of the zooplankton sampling.

The remaining three larger *Metridia* species were predominantly located in the deepest layer (750–1,000 m), both day and night, throughout all seasons. The abundance and biovolume of *Metridia* spp. were observed to be low in February and April but peaked in July and October.

### *Eucalanus bungii*

Figure 5 illustrates the abundance, biovolume, and vertical distribution composition of *E. bungii* during day and night. Unlike *Metridia* spp., which includes multiple species, *E. bungii* consists of only a single species. The mean and standard deviation of the ESD for each developmental stage are presented in Table 2. Consequently, the size of the cohorts in the ESD corresponds directly to each copepodite stage. There were no diel variations in the vertical distribution of *E. bungii* observed in any season. However, a clear seasonal shift in vertical distribution was noted. In October and February, *E. bungii* was found at depths of 200–500 m both day and night. In contrast, during April, especially for the later copepodite stages, the distribution became shallower, occurring in the 50–200 m layer both day and night. In July, the younger copepodite stages, specifically those younger than C4, were located in the shallowest layer of 0–50 m both day and night. The highest abundance of *E. bungii* was recorded in October, with the ESD indicating that the developmental stages dominating this period were C5 and C6.

## DISCUSSION

This study highlights species-specific differences in the vertical distribution and population structure of two income-breeding copepods: *M. pacifica* and *E. bungii*. The main

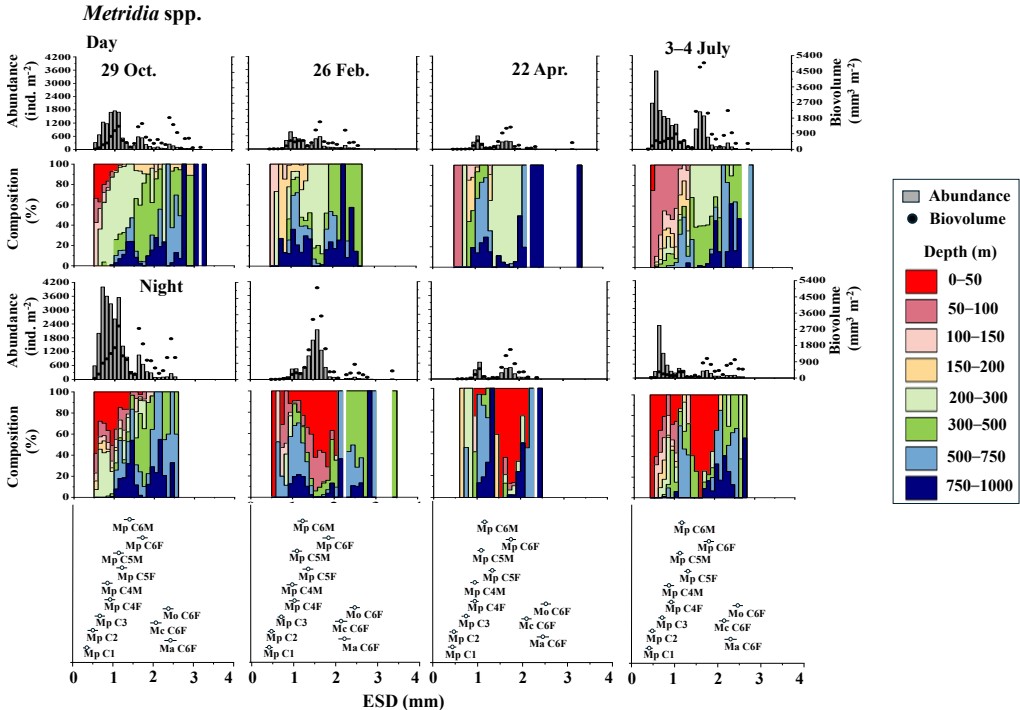

**Figure 4** *Metridia.* spp.: Abundance (ind. m⁻²), biovolume (mm³ m⁻²), and their vertical distribution composition (%) at eight discrete depths along with the equivalent spherical diameter (ESD) at station. For the bottom panel, the mean (symbol) and standard deviation (bar) of the ESD of each stage *Metridia pacifica* and adult female of other *Metridia* species are shown for reference. *M. pacifica* (Mp), *M. okhotensis* (Mo), *M. asymmetrida* (Ma), and *M. curticauda* (Mc). F: female, M: male.

**Table 2  Equivalent spherical diameter (ESD, mm) of each copepodite stage of *Metridia pacifica* and *Eucalanus bungii*.** Values are mean ± 1 sd. F, female; M, male.

| Stage | Metridia pacifica | Eualanus bungii |
|---|---|---|
| C1 | 0.426 ± 0.035 | 0.779 ± 0.066 |
| C2 | 0.517 ± 0.045 | 0.995 ± 0.088 |
| C3 | 0.666 ± 0.051 | 1.475 ± 0.111 |
| C4F | 0.921 ± 0.062 | 2.089 ± 0.116 |
| C4M | 0.872 ± 0.072 | 2.066 ± 0.141 |
| C5F | 1.229 ± 0.076 | 2.865 ± 0.222 |
| C5M | 1.073 ± 0.070 | 2.798 ± 0.187 |
| C6F | 1.726 ± 0.121 | 3.827 ± 0.276 |
| C6M | 1.146 ± 0.061 | 2.971 ± 0.095 |

differences observed are as follows: *M. pacifica* exhibits diel vertical migration (DVM), whereas *E. bungii* follows a pattern of seasonal vertical migration (SVM). Additionally, regarding population structure, *M. pacifica* shows an absence of seasonality, while *E. bungii* demonstrates its presence as a noteworthy characteristic. We will discuss the

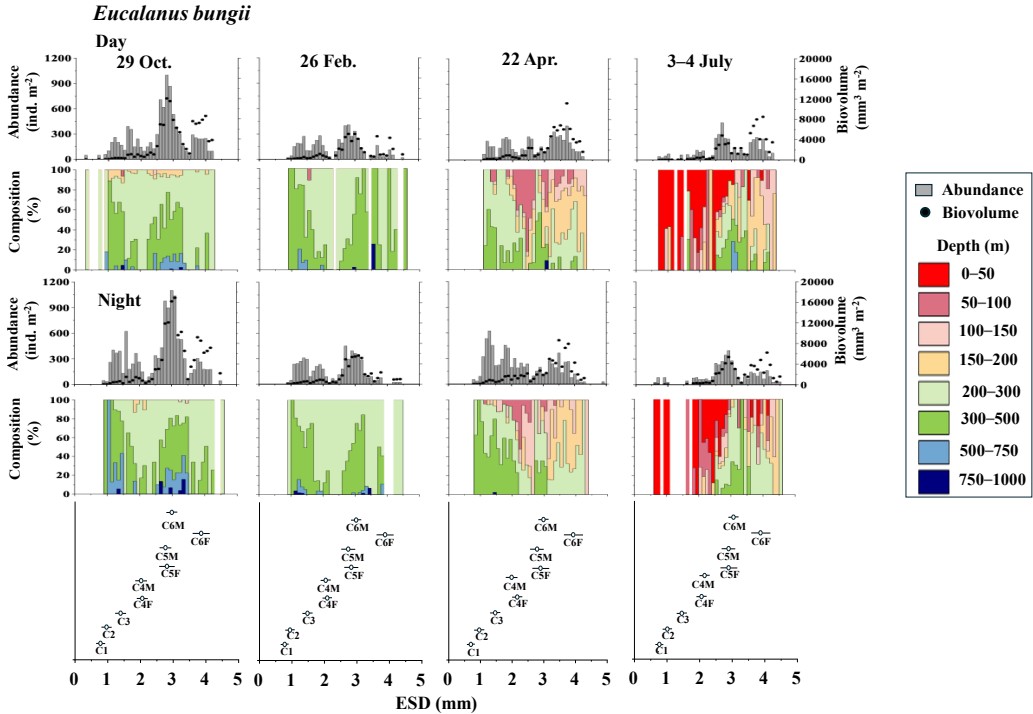

**Figure 5** *Eucalanus bungii*.: Abundance (ind. m⁻²), biovolume (mm³ m⁻²), and their vertical distribution composition (%) at eight discrete depths along with the equivalent spherical diameter (ESD) at sta. For the bottom panel, the mean (symbol) and standard deviation (bar) of the ESD of each copepodite stage of *E. bungii* are shown. F: female, M: male.

ecological features of each species and then make comparisons between these two income-breeding copepods. In conclusion, we will address both the advancements and limitations of ZooScan imaging analysis and suggest future directions for ecological studies focused on specific zooplankton species.

## *Metridia* spp.

Studies conducted at depths of 1,000 m or more in the subarctic Pacific and its marginal seas have reported the presence of seven (*Homma & Yamaguchi, 2010*) or nine (*Yamaguchi et al., 2002*) species within the genus *Metridia*. The most common species is *M. pacifica*, which ranks as the second most abundant species among all calanoid copepods, following *Microcalanus pygmaeus* (*Yamaguchi et al., 2002*; *Homma & Yamaguchi, 2010*). The life cycle of *M. pacifica* has been observed in various locations, including the Gulf of Alaska in the eastern subarctic Pacific (*Batchelder, 1985*), Toyama Bay in the Japan Sea (*Hirakawa & Imamura, 1993*), and the Oyashio region in the western subarctic Pacific (*Padmavati, Ikeda & Yamaguchi, 2004*). The number of generations per year varies by location: in Toyama Bay, only one generation is produced annually (*Hirakawa & Imamura, 1993*), while two generations occur in the Oyashio region (*Padmavati, Ikeda & Yamaguchi, 2004*), and three generations take place in the Gulf of Alaska. In this study, collections occurred only four times over one year (Table 1), which made it challenging to analyze the precise life cycle.

However, during all four observational periods (October, February, April, and July), *M. pacifica* populations exhibited clear diel vertical migration (DVM), ascending to the surface layer at depths of 0–50 m during the night (Fig. 4). While the specific condition of each specimen is known to affect the DVM behavior of *Metridia* species (*Hays, Kennedy & Frost, 2001*), it is commonly known that their C6M remains at the deep layer throughout the day (*Padmavati, Ikeda & Yamaguchi, 2004*). This phenomenon is also well confirmed in this study (Fig. 4). Additionally, *M. pacifica* populations are known to undergo a diapause period, during which they remain in deeper waters and do not migrate to the surface layer both day and night in Toyama Bay and the Oyashio region (*Hirakawa & Imamura, 1993*; *Padmavati, Ikeda & Yamaguchi, 2004*). Based on these observations, it is interpreted that *M. pacifica* in the western subarctic Pacific, where this study was conducted, does not experience a diapause period. Its life cycle appears to resemble that of the population in the Gulf of Alaska, which has repeat generations throughout the year (*Batchelder, 1985*).

Various studies have indicated that the daytime vertical distribution of *M. pacifica* has shifted to a deeper layer as the organisms develop (*Hattori, 1989*; *Sato et al., 2011*). Seasonal changes in the diel vertical migration of *Metridia* spp. are linked to variations in day length throughout the seasons (*Hays, 1995*). In terms of the ontogenetic changes that occur, wherein the daytime distribution depth increases as the copepodite stages progress, *Takahashi et al. (2009)* observed seasonal variations in the daytime vertical distribution of *M. pacifica* and *M. okhotensis* in the Oyashio region. They found that the daytime distribution depth of late copepodite stages (C4–C6F), which occur at depths ranging from 100 to 400 m, is correlated with the depths of the euphotic zone, which lies between 10 and 60 m at the surface. *Takahashi et al. (2009)* clarified that when the euphotic zone is shallow—characterized by abundant surface phytoplankton that blocks light from penetrating deeper—the distribution of late copepodite stages is also shallow. Conversely, when the euphotic zone is deep, indicating a scarcity of surface phytoplankton and greater light penetration into the deep sea, the distribution of late copepodite stages tends to be deeper. They posited that the deeper daytime distribution of larger late copepodite stages is to avoid predation by visual predators, such as pelagic fish. This study also observed that the increase in daytime distribution depth coincides with the later copepodite stages of *M. pacifica* (Fig. 4).

### *Eucalanus bungii*

In studies conducted on pelagic copepods at depths of 1,000 m or more in the subarctic Pacific and its marginal seas, *Eucalanus bungii* is the only species from the genus *Eucalanus* identified in this region (*Yamaguchi et al., 2002*; *Homma & Yamaguchi, 2010*). The size cohorts observed in ESD of *E. bungii* correspond clearly to each copepodite stage, specifically stages C1–C6F/M (Fig. 5). However, from the C4 stage onward, *E. bungii* can be morphologically distinguished between females and males. Males possess a small fifth leg, while females do not (Fig. 2B). Although the body size (ESD) of the C6F stage is notably different from that of C6M, the ESD of females and males overlaps for the C4 and C5 stages (Fig. 5). This overlap makes it challenging to confirm the presence or absence of the small fifth leg in images (Fig. 2B). Additionally, the ESD of small adult C6M

overlaps with that of C5F/M (Fig. 5). Considering these factors, it becomes difficult to accurately identify the copepodite stage of *E. bungii* using ESD alone, particularly for the later copepodite stages.

The life cycle of *E. bungii* has been reported in various regions, including the fjords of British Columbia, Canada (*Krause & Lewis, 1979*), the Gulf of Alaska in the eastern subarctic Pacific (*Miller et al., 1984*), and the Oyashio region in the western subarctic Pacific (*Tsuda, Saito & Kasai, 2004*; *Shoden, Ikeda & Yamaguchi, 2005*). *E. bungii* experiences a deep-sea diapause period during several late copepodite stages, with generation times reported as 1–2 years in the Oyashio region (*Tsuda, Saito & Kasai, 2004*; *Shoden, Ikeda & Yamaguchi, 2005*) and 2–3 years in the Gulf of Alaska (*Miller et al., 1984*). In both regions, the diapause period occurs from October to March of the following year, although there are regional differences in the distribution depths during this period: 250–500 m in the Gulf of Alaska (*Miller et al., 1984*) and deeper than 500 m in the Oyashio region (*Shoden, Ikeda & Yamaguchi, 2005*).

The vertical distribution of *E. bungii* observed in this study showed a concentration at depths of 200–500 m both during the day and night in October and February. Individuals were rarely found below 500 m, with developmental stages C3 and later being dominant (Fig. 5). The vertical distribution of these diapause individuals of *E. bungii* in the western subarctic Pacific is very similar to that of the population in the Gulf of Alaska, as previously mentioned (*Miller et al., 1984*). This consistency aligns with the characteristics of the life cycle of *M. pacifica* in the Gulf of Alaska. The vertical distribution of *E. bungii* exhibited seasonal changes, particularly during the late copepodite stages. In April, individuals were found at shallower depths of 50–200 m both day and night (Fig. 5). This behavioral shift indicates that individuals that were dormant in deeper waters begin to awaken and ascend to shallower layers during the spring phytoplankton bloom period. This phenomenon has been documented in both the Gulf of Alaska (*Miller et al., 1984*) and the Oyashio region (*Yamaguchi et al., 2010a*; *Yamaguchi et al., 2010b*). A short-term time series of daily observations on zooplankton was conducted at one station in the Oyashio region from March to April (*Yamaguchi et al., 2010a*). During this study, vertical stratified sampling was conducted five times over 9-day intervals from depths of 0 to 1,000 m (*Yamaguchi et al., 2010b*). It was noted that from March to April, the gonads of *E. bungii* C6F began to develop, egg production commenced, and nauplii started to recruit (*Yamaguchi et al., 2010a*). Initially, at the beginning of the study, all individuals were located at their diapause depth during both day and night. However, by the second and third sampling periods, they had migrated to the surface, showing no difference in vertical distribution between day and night. This upward migration began first with the late copepodite stage, specifically C6F (*Yamaguchi et al., 2010b*). In April, although the fluorescence values, which indicate phytoplankton abundance, were low, the population of *E. bungii* had started to migrate upward to shallower layers during the late copepodite stage. This upward movement was not observed during the dormant periods in October or February. This suggests that copepods with a multi-year generation length, such as *E. bungii*, may possess an internal clock that prompts them to awaken from dormancy at depth and triggers upward migration (*cf. Mauchline, 1998*).

In July, in this study, all copepodite stages of *E. bungii* (C1–C6F) were present. The early copepodite stages were found in the surface layer, ranging from 0 to 50 m, both during the day and at night (Fig. 5). Adult males (C6M), nauplii, and the early copepodite stages C1 and C2 of *E. bungii* were observed only during a limited seasonal window within a year. In contrast, the diapause developmental stages C3–C6F were present year-round, which is consistent across all the marine areas examined in this study (*Miller et al., 1984*; *Tsuda, Saito & Kasai, 2004*; *Shoden, Ikeda & Yamaguchi, 2005*). This study successfully created a histogram of body length (ESD) for exclusively *E. bungii*, allowing for a clear distinction of the population structure among the early developmental stages (C1–C4F/M) without overlap between the stages. However, distinguishing between males and females in the later developmental stages (C5F/M–C6F/M), and even stage C4, proved challenging. While it was evident that growth and the recruitment of new generation recruits through reproduction occurred in July, evaluating the dynamics of the later copepodite stages and understanding the life cycle, especially generation time, remained difficult using only the image analysis data from ZooScan.

## Inter-species comparison

The two copepod species examined in this study are income breeders that perform reproduction near the surface layer (*Yamaguchi et al., 2010a*). In the western subarctic Pacific, phytoplankton productivity is concentrated in spring, with more than half of the annual production occurring during the spring phytoplankton bloom period (*Ikeda, Shiga & Yamaguchi, 2008*). The population structure of both species was dominated by young copepodite stages in July, just after the spring phytoplankton bloom (Figs. 4 and 5).

All copepodite stages, particularly the younger stages of *E. bungii*, were distributed in the 0–50 m depth layer both day and night in July. A notable ecological difference between the two species is that *M. pacifica* consistently exhibits DVM, while *E. bungii* performs SVM in October and February. During these months, all copepodite stages of *E. bungii* descend to depths below 200 m both day and night. This behavioral difference is believed to be related to the size disparity between the two species, as indicated by their ESD (see Table 2).

While both species exhibit their largest size at the adult females (C6F), *E. bungii* is 2.22 times larger than *M. pacifica*. This size difference translates into a biovolume difference of approximately 10.9 times (*i.e.*, $2.22^3$). For the larger-bodied *E. bungii*, reaching C6F in one year presents challenges, as it is reported to require multiple years to reach this stage in both the eastern and western subarctic Pacific (*Miller et al., 1984*; *Shoden, Ikeda & Yamaguchi, 2005*). To accommodate such a prolonged generation length, *E. bungii* enters a dormant phase during which it descends to deeper layers, remaining at that depth throughout the day during seasons of reduced primary production at the surface, ceasing to feed and develop (*Tsuda, Saito & Kasai, 2004*; *Tsuda, Saito & Kasai, 2014*; *Tsuda et al., 2015*).

In contrast, *M. pacifica*, which has a smaller body size, requires significantly less energy to grow—approximately one-tenth of that required by *E. bungii*. This energy advantage allows *M. pacifica* to grow without a resting phase throughout the year. Consequently, this life cycle pattern enables *M. pacifica* to continue performing DVM even during seasons when their large-body size competitors, *E. bungii* and *Neocalanus* spp., descend to deeper

waters for resting. *M. pacifica* can remain near the surface to feed and may reproduce throughout the year by exploiting the phytoplankton niche left vacant by larger species that dive into deeper waters (*Batchelder, 1985*; *Padmavati, Ikeda & Yamaguchi, 2004*).

In summary, while both copepod species in this study are income breeders that feed on particles, clear interspecific differences exist in their life cycles. The primary factor influencing these differences is body size and mass, as evidenced by the 2.22 times difference in ESD (Table 2), which results in a notable 10.9 times difference in biovolume. This disparity in size and biovolume directly affects the energy requirements for maturation and growth in these species.

## Advantages, shortcomings, and future perspectives on ZooScan

In this study, we utilized the imaging device ZooScan to analyze the vertical distribution and population structure of two dominant planktonic copepods. ZooScan is commonly employed to assess the size composition of the entire zooplankton community and to monitor geographical and seasonal changes in normalized biomass size spectra (NBSS) (*cf. Naito et al., 2019*; *Teraoka et al., 2022*, and references therein). On the other hand, *Taniguchi et al. (2023)* used imaging data collected by ZooScan to study amphipods. They sorted all amphipods from the same IONESS net sample used in this study and performed ZooScan measurements on the sorted amphipod samples. Species identification was conducted manually based on the obtained images, enabling accurate differentiation between males, females, and juveniles of the dominant species (*Themisto pacifica*). Additionally, as the equivalent spherical diameter (ESD) is regarded as an index of body size, the population structure of the dominant species could be accurately determined, and their cohort analyses were made using the number of 93-1,138 individuals (mean: 601 individuals and 4,807 individuals for the whole seasonal and day/night samples, *cf.* Figure 9 of *Taniguchi et al., 2023*).

One advantage of ZooScan is its ability to rapidly obtain precise body size data for a large number of individuals while minimizing personal bias. In contrast, this study analyzed unsorted whole zooplankton samples using ZooScan, requiring manual species identification based on the captured images. Identifying species from image data for the *Neocalanus* genus, which comprises three congeneric species, is particularly challenging, so we excluded them from our analysis. Moreover, the late copepodite stages of *E. bungii* are classified as dormant (resting) individuals based on the presence or absence of food particles in their digestive tract (gut), indicating whether they are actively feeding or resting (*Tsuda, Saito & Kasai, 2004*; *Tsuda, Saito & Kasai, 2014*; *Tsuda et al., 2015*). While such separation is necessary for identifying diapause in late copepodite stages of *E. bungii*, performing this analysis with image data can be difficult.

ZooScan has the advantage of quickly obtaining a substantial amount of precise area data (*Gorsky et al., 2010*; *Irisson et al., 2022*). When using a standard stereomicroscope, the major and minor axes are measured, and volume is calculated using a geometric shape formula (*Gorsky et al., 2010*). For example, while measuring the volume of the prosomes of copepods can be done accurately, separate measurements of the urosome are also necessary for a complete quantification of total volume (= prosome volume + urosome

volume), which can be time-consuming (*Mauchline, 1998*). Although imaging analysis using a stereomicroscope is feasible, it is not a commonly employed method.

Future research will likely capitalize on ZooScan's advantages by estimating zooplankton fluxes (feeding, production, and egestion). ZooScan can easily obtain accurate "Area Excluded" data, which can then be converted to dry weight using species-specific equations (*Lehette & Hernández-León, 2009*; *Maas et al., 2021*). This allows for the estimation of zooplankton respiration (oxygen consumption rate) from a "global-bathymetric model" that incorporates four independent variables: zooplankton dry weight, habitat temperature, depth, and zooplankton taxa, which can be estimated at a precision of 93.4–94.2% (*Ikeda, 2014*). Consequently, we can estimate their associated fluxes (feeding, production, and egestion) within energy budget models that consider specific dynamic action, which varies across zooplankton taxa (*Ikeda, 2021*). Therefore, we envision future directions for zooplankton imaging studies to focus on accurately estimating fluxes using the imaging data obtained.

## ACKNOWLEDGEMENTS

We would like to express our heartfelt gratitude to Dr. Minoru Kitamura of the Japan Agency for Marine-Earth Science and Technology (JAMSTEC) for collecting and providing the IONESS net samples used in this study, as well as for his valuable suggestions. We also sincerely thank all the researchers involved in the project, including Dr. Makio Honda of JAMSTEC, who served as the K2S1 research project leader. Our thanks also go to the captain and crew of the JAMSTEC research vessel Mirai for their efforts in conducting the offshore work. We also thank the World Association of Copepodologists for being a Hub for PeerJ of this article.

### Funding

This work was supported by the Arctic Challenge for Sustainability 3 (ArCS-3) Program (Grant Number JPMXD1720251001) from Ministry of Education, Culture, Sports, Science and Technology (MEXT), Grants-in-Aid for Challenging Research (JP20K20573 [Pioneering] to Atsushi Yamaguchi), and Scientific Research (JP22H00374 [A] to Atsushi Yamaguchi) from the Japanese Society for the Promotion of Science (JSPS). The funders had no role in study design, data collection and analysis, decision to publish, or preparation of the manuscript.

### Grant Disclosures

The following grant information was disclosed by the authors:
Ministry of Education, Culture, Sports, Science and Technology (MEXT): JP-MXD1720251001.
Grants-in-Aid for Challenging Research: JP20K20573.
Japanese Society for the Promotion of Science (JSPS): JP22H00374.

## Competing Interests

The authors declare there are no competing interests.

## Author Contributions

- Tian Gao conceived and designed the experiments, performed the experiments, analyzed the data, prepared figures and/or tables, authored or reviewed drafts of the article, and approved the final draft.
- Atsushi Yamaguchi conceived and designed the experiments, analyzed the data, prepared figures and/or tables, authored or reviewed drafts of the article, and approved the final draft.

## Data Availability

Raw data is available in the Supplemental Files.

## Supplemental Information

Supplemental information for this article can be found online at http://dx.doi.org/10.7717/peerj.19956#supplemental-information.

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
