# Peer review of "Analysis of imaging data on seasonal changes in the population structure and vertical distribution of two dominant planktonic copepod species in the western subarctic Pacific"

_PeerJ, doi:10.7717/peerj.19956_

## Round 0.1 · original submission · Major Revisions

There is disagreement among the three reviews. The first two suggest changes to the writing of the manuscript and emphasize how you can make the paper more relevant (and citable!) to the reader. The third review, however, finds the manuscript unsuitable for publication in its current form. This review is thorough and thoughtful, identifying numerous issues that need to be addressed and providing guidance on how to address them. You may not agree with all of the issues raised by the reviewer, but you should consider them and state why or why not you follow the reviewer's suggestions.

·

Basic reporting

Good

Experimental design

Good

Validity of the findings

Good

Additional comments

Here zooplankton are collected from different depths in the Pacific using net samples and then identified and sized by an electronic system, ZooScan. The vertical distribution of different dominant species is then described and seasonal patterns discussed.

This is a solid manuscript with some nice results. Some clear patterns are identified. At the moment the writing is rather parochial – centred around this particular study site. I think this will limit impact of this work. I think that the authors could be a little more expansive in their considerations – there is a lot known about vertical migration in these genera that is not mentioned. I have a few suggestions for minor revisions. With a little care, I think these changes will be straightforward.

1. Line 107. Explain if a flowmeter was used to measure the volume of water filtered.
If there was not a flowmeter, how would variable net clogging impact your results ?

2. Line 107. “… we conducted day and night vertically stratified oblique tows ,,,”
Be a little more specific about what you mean by “day” and “night” in this regard. Midnight and midday + /- X hours ?

3. Line 236. “Additionally, M. pacifica populations are known to undergo a diapause period, during which they remain in deeper waters and do not migrate to the surface layer both day and night in Toyama Bay and the Oyashio region …”
Note that studies with M. pacifica and some other migrating zooplankton have shown that sometimes only a proportion of individuals migrate to the surface – those in good body condition might sometimes not migrate. See: “Individual variability in diel vertical migration of a marine copepod: why some individuals remain at depth when others migrate. Limnology & Oceanography 46, 2050-2054 (2004)” and subsequent papers. Can you explain whether a proportion of the population did not migrate ? This appears to be the case looking at Figure 4.

4. Takahashi et al. (2009) observed seasonal variations in the daytime vertical distribution of M. pacifica and M. okhotensis in the Oyashio region …”
I would be a little more expansive in your writing here. It is well described for Metrdia how the length of seasonal near-surface occupation each night varies with the daylength – animals spend longer at the surface in the winter when nights are longer. See for example: “Hays GC (1995). Ontogenetic and seasonal variation in the diel vertical migration of the copepods Metridia lucens and Metridia longa. Limnology and Oceanography 40, 1461-1465. (2005)”.
So I would discuss that there are likely seasonal changes in the timing of DVM as well as the amplitude of DVM.

5. The term amplitude of DVM is often used to describe the difference between day and night depths. The term is not mentioned here. I would describe the amplitude of DVM and place in the broader context of what is known elsewhere. There is lots known about many species of copepods, including Metridia but also Pleuromamma. I would try and mention this broader literature and how your findings fit in.

In summary, a nice solid manuscript that I look forward to seeing published. Graeme Hays

·

Basic reporting

No comment

Experimental design

No comment

Validity of the findings

No comment

Additional comments

I have marked the suggested revisions and comments within the manuscript file for the authors' consideration.

Reviewer 3 ·

Basic reporting

- Language and structure
The present manuscript aims at describing the structure of population, vertical distribution and growth of two copepod species through taxonomical analysis and image analysis. The manuscript is written in a perfectly clear English, and the structure of the manuscript is well highlighted and conforms to PeerJ standards.
- References and background
In general, relevant references and general knowledge are provided to explain the points brought by the authors or to explain what other studies have been made on similar subjects. However, some analyses are done without explaining their usefulness or ecological/biological relevance. For example, it is not obvious why it is interesting to measure intermolt growth calculation, or to know which species of Metridia spp. is migrating vertically or not. In my opinion, background it is not enough to understand the relevance of the overall study and research questions are not clearly stated (see 2. Experimental Design).
- Figure and tables
Figures and tables are of good quality and well labeled. Note that the legend is in inserted below the figure and should probably be removed for publication. Figures are easily readable; however, for the most complex figures (Fig. 4 and 5), the associated text in the Result part could better explained to help the reader following and understanding it. Also, some figures’ content is not relevant, or not used in the text.
In details:
o Figure 1 is not cited in the text, and the information found in it such as currents circulation is quite complex but not used at all in the manuscript and interpretations, so I am not sure it is relevant.
o Figure 2 is really good and help to understand population of the two copepods species
o Figure 3 is clear (there is no unit for Fluorescence?)
o Figure 4 and 5 are interesting, but complex so need to be better described in the text for example say that upper and lower panels should be compared and if more red colors are seen, it means that these copepods are more at the surface). Then, the last panel with ESD per species and stages could be lightly separated from the other, showing that this panel help to interpret either day or night variations.
o Figure 6 is good as a figure; however, I do not understand why it is useful to calculate ESD and Biovolume growth (see comments hereafter).
o Figure 7 is not clear at all, I would recommend a growth curve with points and standard deviation to have an idea of the size and the growth, here it is hard to interpret and lead to some mistakes (we could think that you could differentiate males and females at young stages, which is not the case).


- Raw data
I thank the authors for providing the raw data. However, as data is provided through individual excel sheets (9 different seasons and species with 16 sheets inside each so more than 140 sheets in total, in addition to sheets on ESD and biovolumes), it makes it really difficult to use and understand for readers. Columns and rows are formatted differently, and this descriptive metadata would be necessary to understand the datasets. There is no code, neither excel workflows or other information to explain how the figures and analyses are produced. Finally, if images are processed under the web platform Ecotaxa, it should be easy to make them accessible or at least visible (Irisson et al. 2022). For all these reasons, the reproducibility standards of PeerJ does not seem respected.

- Self-contained study
The study is self-contained, but no clear hypotheses are formulated.

Experimental design

- Original primary research
This study is an original primary research, and by describing the biology of zooplankton populations in 4 seasons, it fits in the Scope of PeerJ journal for environmental and biological studies.
- Knowledge gaps
A lot of work has been done for sampling, taxonomical identification of species and stages and for the use of the Zooscan tool and Ecotaxa web platform. But the main weakness of the study is that research questions are not well defined, and no biological or ecological hypotheses are formulated. We understand well how are examined population structure, vertical distribution and growth of these two copepods, but we do not really understand where are research gaps and what is really new or brought by this study. Just by reading the abstract, we understand what are the results, but not the implications. Perhaps the main aim is to understand the biology of poorly known species, but even this aspect is not clearly mentioned; and knowledge gaps about growth are not mentioned at all.

- Technical and ethical standards
This work appears rigorous and performed with good technical and ethical standards. The weakness is mostly due to a lack of research questions and the relevance of the results and interpretations.

- Methods details and reproducibility
Methods are quite well described and, for most of them, could be reproduced with another dataset (but not with these raw data as their format is too confusing). An important exception is for the biovolume calculation, that should be explained somewhere.

Validity of the findings

Some findings presented in this study are likely adding knowledge concerning biology of the two species at sampling location, but these implications are not are not detailed adequately. Then, the findings about intermolt growth are not robust enough, and to my opinion, do not add significant knowledge about copepod’s biology.

In details:
- One part of the study aims at understanding the population’s structures of two dominant copepod species Metridia longa and Eucalanus bungii through the distribution of stages and/or size classes. I do not have extensive knowledge on these 2 species and this region, but I understand that these research results can be valuable if the species are not well known, to better understand ecosystem functioning and knowledge about biodiversity. However, these last points are not mentioned and the study of the species mostly appear opportunistic. More justifications are needed.

- Another goal of the study is to explain vertical distribution by species and stages and/or size class and season. These results and associated figures 4 and 5 are interesting and likely add important knowledge for these species. But once again, better explanations are needed to understand what new knowledge, or new confirmation of known biological processes, is given by this study.

- To my opinion, results about intermolt growth should be reconsidered because they do not seem robust. I am not really familiar with this measurement, but in the introduction, we do not understand the ecological relevance of studying this aspect despite it could probably be better justified. Then, the main result is that “biovolume growth is significantly higher than ESD growth”, which do not seem as a strong result. Indeed, calculation of the biovolume is not mentioned anywhere but I assume that it is calculated from the ESD in mm and converting it to a volume, likely considering the copepod as a sphere or an oval (examples in Drago et al. 2022 10.3389/fmars.2022.894372 for another imaging system). If this is correct, it is straightforward that biovolume growth is higher than ESD growth, because of the power three. The allometric relationship between size and volume is simply revealed (ref Lombard) but I do not understand how this could be considered as a new result. I would argue that the use of the biovolume to compute growth can be sufficient (because animals grow in three dimensions) and could be used to compare growth between stages or between males and females, which would be the main result to discuss (and not the fact that biovolume growth is more important than ESD growth).
Regarding male and female growth comparison, the method explained in “Additionally, the proportion of growth at each copepodite stage was assessed for both body length (ESD) and volume (biovolume), with the values for adult males and females set at 100% (Yamaguchi et al., 2020) (l. 159)” is not clear. If I understood well, you normalized the growth of each copepodit by the adult size of males or females and as females are larger than males at the end of the growth (when it is possible to assign them as male or female), it looks quite obvious than larger growth appears for female at late stages in comparison with males. So, the main result would be that females are larger than males, and I think that it is quite well known for copepods. As explained above, a growth curve would be easier to interpret and then, it would be more interesting to compare species between them or to evaluate if it is well known that females are larger, or grow later than males for other copepod species. Again, what is new or unique in your result in comparison with other studies?

- More generally, the use of the Zooscan is often mentioned as an argument for the originality and relevance of the study. The use of a novel instrument is not enough to justify the validity of a study and it should be better justify why using the Zooscan adds something to the study. For example, the sentence “The advantages of ZooScan for in-depth analysis of specific planktonic taxa are increasingly clear” (l. 70) is not detailed enough.
One argument you mentioned is that Zooscan allows the calculation of ESD and biovolumes. This is true, but this information was also possible to quantify before the use of the Zooscan, through individual measurements of individuals under the stereomicroscope. The use of the Zooscan probably allows quicker analysis and thus higher number of individuals measures. It is also likely lead to more consistent result as there less human bias in the measurement of the size. These could be arguments (and other advantages might be found) but they have to be justified.

- Underlaying data and statistics
Interpretations are only based on the results presented in this study in a descriptive manner. No statistical analysis is performed. They are not always absolutely necessary as some parts are more descriptive (i.e. vertical distribution of species). But especially for growth calculations and comparisons, knowing average size with standard deviations; and clear statistical comparisons, would be useful. Adding statistics could be beneficial to the manuscript, but this is a secondary priority after the definition of clear research questions and the justification on why some metrics (i.e. intermolt growth) are used.

- Conclusions linked to original research question and limited to the results
As said previously, results are explained in a clear manner, but interpretation and conclusions are difficult to link with any research question because they are not well stated in the introduction.

Additional comments

The main things that would need to be change in order to improve the manuscript:
1. Introduction needs to be refined, especially with the restructuration of the 3 middle paragraphs to have one idea per paragraph leading to the research question and hypothesis
2. Less emphasis should be made on the use of the Zooscan, but more on biological and ecological questions, and only justify how the Zooscan can help answering them in a new/better/quicker way.
3. Calculation of intermolt growth have to be justified from literature background; biovolume vs ESD analysis should be reconsidered as it is maybe not new result
4. Comparison between male and female intermolt growth should be done with a growth curve and statistical analyses; and then put in a better literature context (is it just caused by the fact that females are larger than males at the end of their life cycle?).

Details per lines:
l. 43: Station K2 is mentioned as a “long-term time-series observation station” but then for the study you only have 4 points on 2 years. Could you explain how long is this time serie and why only a portion can be applied to your study?

l. 56: I don’t understand how it is possible that many zooplankton species were studied but that “population structure of dominant species”, such as Metridia and Eucalanus, could still be a knowledge gap.

l. 61: first paragraph explains all the science done at station K2. It is interesting as it shows the good ecological background from the literature, but it looks like an enumeration but miss strong statements about the usefulness of the study.

l.68: what details about abundance and biovolume are accessible by the zooscan and not by stereomicroscope studies? I feel that the difference is mostly about processing time and not data quality.

l. 72: again, if “copepods are the most dominant group in term of abundance and biomass”, why are their populations dynamics not well studied in comparison with amphipods, chaetognaths, etc.

l.90: “image analysis can be performed on samples sorted by copepodite stage”: so you sorted taxonomy before putting samples in the Zoocan? Do you use Zooscan and Ecotaxa as a classifier or just as a way to measure individuals? This would need to be precised in the methods.

l. 91: much more details are needed avoid intermolt growth

l.94: The 3 last paragraphs above this line move back and forth between different ideas, often ended by a key point on imagery. They could be merged or restructured to have one idea per paragraph, then leading to a clear research questions with hypotheses.

l. 117: the part on Zooscan should be put after Data analysis if taxonomical sampling was done before Zooscan use.

l. 139: the term “standing stock” is not often used to my knowledge, why are you using it?

l. 155: calculation of biovolume needed
l. 159: intermolt growth per sex calculation is not clear

l.205: “Thus, it is evident that inter-molt growth based on biovolume was greatly higher than that based on ESD », to me, this is a not a new result

l. 209: Figure 7 is not clear neither statistically robust

I will not provide detailed comments on the discussion because the other comment will induce changes in the discussion.

---

## Round 0.2 · accepted · Accept

Dear Tian Gao,

I have now received two evaluations of your revised manuscript from previous reviewers. I am satisfied that you addressed the reviewers' concerns and suggestions in your revised manuscript. I am recommending that the paper be accepted now.

Thanks for submitting your work to the WAC hub at PeerJ.

Best wishes,

Hans Dam
Associate Editor, WAC Hub

·

Basic reporting

The authors have made a good effort to revise the manuscript in line with the comments. I think this manuscript can now be accepted for publication. It will make a nice contribution. Graeme Hays

Experimental design

The authors have made a good effort to revise the manuscript in line with the comments. I think this manuscript can now be accepted for publication. It will make a nice contribution. Graeme Hays

Validity of the findings

The authors have made a good effort to revise the manuscript in line with the comments. I think this manuscript can now be accepted for publication. It will make a nice contribution. Graeme Hays

·

Basic reporting

No comment

Experimental design

No comment

Validity of the findings

No comment

Additional comments

The authors have made the necessary corrections and provided detailed explanations in line with the suggestions I made in my previous review. I thank the authors for considering my contributions.